# Polycystic Kidney Disease Ryanodine Receptor Domain (PKDRR) Proteins in Oomycetes

**DOI:** 10.3390/pathogens9070577

**Published:** 2020-07-16

**Authors:** Limian Zheng, Barbara Doyle Prestwich, Patrick T. Harrison, John J. Mackrill

**Affiliations:** 1Department of Physiology, School of Medicine, University College Cork, T12 XF62 Cork, Ireland; 107139861@umail.ucc.ie (L.Z.); p.harrison@ucc.ie (P.T.H.); 2School of Biological, Earth and Environmental Sciences, University College Cork, T23 TK30 Cork, Ireland; b.doyle@ucc.ie

**Keywords:** oomycete, *Phytophthora infestans*, calcium, polycystic kidney disease channel, inositol 1,4,5-trisphosphate receptor, ryanodine receptor, evolution, biochemistry

## Abstract

In eukaryotes, two sources of Ca^2+^ are accessed to allow rapid changes in the cytosolic levels of this second messenger: the extracellular medium and intracellular Ca^2+^ stores, such as the endoplasmic reticulum. One class of channel that permits Ca^2+^ entry is the transient receptor potential (TRP) superfamily, including the polycystic kidney disease (PKD) proteins, or polycystins. Channels that release Ca^2+^ from intracellular stores include the inositol 1,4,5-trisphosphate/ryanodine receptor (ITPR/RyR) superfamily. Here, we characterise a family of proteins that are only encoded by oomycete genomes, that we have named PKDRR, since they share domains with both PKD and RyR channels. We provide evidence that these proteins belong to the TRP superfamily and are distinct from the ITPR/RyR superfamily in terms of their evolutionary relationships, protein domain architectures and predicted ion channel structures. We also demonstrate that a hypothetical PKDRR protein from *Phytophthora infestans* is produced by this organism, is located in the cell-surface membrane and forms multimeric protein complexes. Efforts to functionally characterise this protein in a heterologous expression system were unsuccessful but support a cell-surface localisation. These PKDRR proteins represent potential targets for the development of new “fungicides”, since they are of a distinctive structure that is only found in oomycetes and not in any other cellular organisms.

## 1. Introduction

Oomycetes superficially resemble fungi in appearance, but phylogenetically belong to the stramenopile (or heterokont) lineage, that includes brown algae and diatoms [1]. *Phytophthora infestans* is an exemplar oomycete that has been a focus of sustained investigation owing to its devastating global impact on potato and tomato production, as the causative organism of late blight [2]. Other oomycetes are pathogens of a wide range of hosts, causing economic losses in the agriculture, horticulture, forestry and fisheries sectors. These losses are not only due to the direct effects of disease, but also the expense of applying protective measures, such as spraying crops with anti-oomycete “fungicides”. Many anti-oomycete pesticides are becoming less effective, owing to either legislative changes banning their use, or due to emergent resistance in oomycete populations. For example, some naturally occurring strains of *Phytophthora capsici* (causing stem and root rot of *Capsicum spp.*) are resistant to the recently developed compound oxathiapiprolin [3]. Improved insights into the biology of *P. infestans* and other oomycetes might lead to the development of new strategies for combatting these pathogens.

As with all cellular organisms [4], intracellular ionised free Ca^2+^ is a key second messenger in the control of oomycete physiology. Cells utilise a toolkit of Ca^2+^-regulating proteins, including channels, transporters (pumps and exchangers), buffers and effectors to control and to respond to this second messenger [5]. In most cells, ionised free Ca^2+^ is found at millimolar concentrations in the extracellular environment, compared with nanomolar to micromolar levels in the cytoplasm. Extracellular and intrinsic stimuli alter the gating of Ca^2+^ channels, increasing cytoplasmic levels of this ion. Together with transporters and buffers, this generates distinctive spatiotemporal patterns of intracellular Ca^2+^, altering the activities of particular subsets of effector proteins, leading to specific cellular outcomes [6]. Ca^2+^ channels in the cell-surface membrane can be gated in response to a variety of cues, such as changes in membrane potential, temperature, ligand binding, mechanical stimuli, or combinations of these. In contrast, Ca^2+^ channels residing in the membranes of intracellular organelles such as the endoplasmic reticulum (ER), vacuoles, lysoendosomal system or mitochondria must be coupled to extracellular cues via second messengers or by direct, allosteric interactions with partners residing in the surface membrane. In eukaryotes, certain G-protein-coupled receptors can activate phospholipase C enzymes that hydrolyse the lipid phosphatidylinositol 4,5-bisphosphate to generate the second messenger inositol 1,4,5-trisphosphate (IP_3_). This second messenger binds to and gates ER Ca^2+^ channels called the IP_3_ receptors (ITPRs), allowing the release of Ca^2+^ into the cytoplasm [7,8]. A family of Ca^2+^-release channels called the ryanodine receptors (RyRs) are distantly related to the ITPRs and are found in metazoans (animals) and their sister groups, such as choanoflagellates and capsasporans [9,10]. RyRs are gated by mechanical interaction with voltage-gated Ca^2+^ channels in the cell-surface membrane, or by Ca^2+^ itself in an amplification process termed Ca^2+^-induced Ca^2+^-release [11].

Cytoplasmic Ca^2+^ plays critical roles in controlling oomycete physiology [12]. Zoospores are a motile stage of oomycetes, which contribute to the propagation of disease. Modification of the concentration of Ca^2+^ in the extracellular environment, or application of the voltage-gated Ca^2+^ channel blocker amiloride, modifies the swimming patterns of several *Pythium* species [13]. Knockdown to the heterotrimeric G-protein α-subunit in *Phytophthora sojae* decreases chemotaxis of zoospores towards the plant isoflavone daidzein, indicating a role for the phospholipase C-IP_3_-ITPR-Ca^2+^-signalling pathway in this process [14]. The L-type voltage-gated Ca^2+^ channel antagonist nifedipine is reported to inhibit the mycelial growth and sporulation of *P. capsici* [15].

As measured using a pressure-injected Ca^2+^-sensing fluorescent dye, fura-2-dextran, the resting free Ca^2+^ concentration of the sporangia of *P. cinnamomi* is in the order of 100 nM, consistent with that in many eukaryotic cell types [5]. Within a minute of cold shock, a stimulus that promotes zoosporogenesis and zoospore release, there was a transient rise in cytoplasmic Ca^2+^, followed by a sustained increase in the level of this second messenger. Microinjection of the Ca^2+^ chelator 5,5′-dibromo-BAPTA prior to cold shock abolished the transient rise in cytoplasmic Ca^2+^ and also inhibited subsequent cytokinesis [16]. Mating pheromone-induced death 1 (MID1) protein is a component of a Ca^2+^ channel in the yeast *Saccharomyces cerevisiae*. Gene silencing of a homologue of this channel subunit in *P. parasitica* prevented cold shock-induced zoosporogenesis. This effect could be partially rescued by increasing the concentration of extracellular Ca^2+^ [17]. Analysis of changes in mRNA expression involved in *P. infestans* zoosporogenesis revealed significant upregulation of 15 genes. For most of these genes, pharmacological inhibitors of voltage-gated Ca^2+^ channels (verapamil), or of the IP_3_-pathway (U73221, a phospholipase C antagonist; 2-aminoethoxydiphenyl borate, an inhibitor of ITPR gating) abolished these increases in transcription [18]. RNA-seq analysis of *P. infestans* highlighted the potential involvement of Ca^2+^ signalling during the life cycle of this organism. In particular, the transcription of several genes encoding Ca^2+^-dependent protein kinases (Ca^2+^ effectors) increased during zoosporogenesis. Similarly, genes for two polycystin-2/polycystic kidney disease-2 (PKD2) homologues, related to mammalian transient receptor potential (TRP) superfamily of Ca^2+^-conducting cation channels, were also upregulated during zoospore formation [19].

Since oomycetes possess extensive endomembrane systems, including the ER and vacuolar membranes [20], we have sought to characterise Ca^2+^ channels that reside in these organelles. Previously, we identified candidate homologues of mammalian ITPR and RyR channels in oomycetes, including *P. infestans* [10,12]. The candidate ITPR homologues varied in number between different oomycete families and some did not appear to contain either consensus IP_3_ binding (I) (a table of acronyms for calcium channels and their constituent domains is provided in Appendix A) or ion transport (IT) domains, so might not function as IP_3_-gated Ca^2+^-release channels. One aim of the current study was to clarify the phylogenetic relationships and potential functions of putative oomycete ITPRs. In addition, we identified a family candidate oomycete Ca^2+^-release channels that displayed strong identity with mammalian polycystic kidney disease (PKD) cation channels and also contained two RyR domains (R), that are a characteristic feature of metazoan RyRs. We coined these channels PKDRR proteins, although they are often annotated as ryanodine-inositol 1,4,5-triphosphate receptor Ca^2+^ channel proteins in oomycete genome databases. However, PKDRR channels are currently hypothetical, having been identified via conceptual translation of oomycete genomes. Furthermore, mammalian PKD proteins typically reside in the cell-surface membrane, whereas ITPRs and RyRs are usually located in ER-derived membranes. Consequently, in the current study, we sought to verify whether PKDRR proteins are actually produced in *P. infestans*, to investigate their subcellular localisation(s), to reconstruct their evolutionary relationships, and to characterise their biochemical and functional properties.

## 2. Results and Discussion

### 2.1. Oomycetes Genomes Encode Putative ITPR Homologues

Although ITPRs are employed widely in eukaryotes as a pathway for the release of Ca^2+^ from organelles [9,10], little is known about their structure(s) and role(s) in oomycetes [12]. In addition, in oomycete proteomes, the PKDRR proteins that are the focus of the current work are frequently annotated as belonging to the ITPR/RyR superfamily of channels (also called the “ryanodine-inositol receptor-calcium channels” superfamily). Consequently, we sought to identify putative oomycete ITPR homologues by searching conceptual translations of oomycete genomes with the protein sequence of human type 1 inositol 1,4,5-trisphosphate receptor (ITPR1), using the basic local alignment of sequences tool (BLAST) at the Pubmed website: https://blast.ncbi.nlm.nih.gov/Blast.cgi?PROGRAM=blastp&PAGE_TYPE=BlastSearch&LINK_LOC=blasthome/.

This revealed that most oomycete proteomes contain at least one ITPR homologue, as summarised in Figure 1 and in Appendix A. For example, *P. infestans* encodes one putative ITPR homologue, whereas *Saprolegnia diclina* (an oomycete that causes cotton moulds in fish) has four predicted members of this superfamily. This suggests expansion of the ITPR family in some oomycete taxa. In reconstruction of the evolutionary history of ITPRs, generated using the maximum likelihood method on MEGA-X software [21], oomycete members of this superfamily fall into two major groupings, termed “Family A” and “Family B”. These oomycete ITPR families do not cluster with those from other taxa, including groups of RyRs, ITPRs from metazoans (animals) and sister groups (fungi and related taxa), and those from cilophorans (multinucleate protozoans), as shown in Figure 1A. This suggests that ITPRs evolved early during eukaryotic evolution, prior to the divergence of the major eukaryotic groups [9,10]. This pre-dates the origins of oomycetes, which are estimated to have emerged in the mid-Paleozoic (430–400 Ma), at the latest [22].

Protein domains are structurally distinct components within proteins, typically greater than 100 amino acid residues in length, that often have defined functions. Assembly of different combinations of protein domains into different architectures defines the function of each specific protein. Rearrangement of protein domains into new combinations is one mechanism by which novel protein functions evolve [23]. One strategy for identifying protein domains uses multiple sequence alignments (MSA) to identify conserved features within protein families. This approach resulted in the development of NCBI’s conserved domain database (CDD) [24]. We characterised the domain architectures of ITPR homologues identified in this study using this tool, Appendix A. Some of these architectures are represented in Figure 1B. As described previously, metazoan ITPRs and RyRs possess related protein architectures, sharing multiple domains in the same sequence along the protein [10]. These include an ITPR (I) domain, thought to be involved in coupling ligand binding to channel gating [8]; a mannosyl transferase, ITPR and RyR (MIR) domain [25]; an RyR and ITPR homology (RIH) domain; a RIH associated (RIHA) domain; and the IT channel domain [7].

Some oomycete ITPR homologues have similar protein domain architectures to metazoan ITPRs. Others either appear to lack certain domains or have additional domains within their structures. For example, the single ITPR homologue of *P. infestans* has a similar organisation to that of *H. sapiens* ITPR1, but contains an additional “Type IV secretory pathway, VirB10 component” domain located near to its N-terminus. Proteins from *S. declina* and *Achyla hypogyna* (a saprolegnialean oomycete) contain I, RIH, RIHA and IT domains, along with an ankyrin repeat domain near their N-termini. Conversely, an IT domain could not be detected in multiple oomycete ITPR homologues. This suggests that these homologues might not function as ITPR-like cation channels, or that the software could not detect their ion channel domains.

The RyR family of Ca^2+^-release channels contain several domains that are absent from ITPR architectures. One of these is termed the RyR domain (R) and this was only detected in eukaryotic RyRs (most of which have four of these domains); in several families of oomycete proteins; and in multiple bacterial, archaeal and viral proteins [10], as shown in Figure 1B. The function of the R domain is unknown. In earlier studies, we identified the R domain in a family of hypothetical oomycete proteins (deduced from conceptual translation of genomes) that we called the polycystic kidney disease (PKD)-R-R channels [10,12]. In genome and proteome databases, members of this PKDRR family are sometimes referred to as “ryanodine-inositol receptor-calcium channels”. We consider this to be a mis-annotation since this candidate channel family shares no strong identity with ITPRs and is only related to RyRs through the possession of R domains. PKDRR channels are a family of proteins that are only detectable in oomycetes and are the focus of the current study.

### 2.2. Characterisation of the PKDRR Family of Proteins

Previously, we have proposed that PKDRR proteins are an oomycete-specific family of cation channels [10,12]. To gain insights into the evolution of this family, we constructed a phylogenetic tree and analysed protein domain architectures using data from eukaryotic taxa that were not available in our past work, as shown in Figure 2. These approaches indicate that PKDRR channels belong to the PKD family of cation channels proteins, which in turn belong to the TRP superfamily [26]. Our analyses also demonstrate that there are two distinct subfamilies of PKDRR proteins: PKDRR A, which have a C-terminus that extends more than 10 amino acid residues from the second R domain; and PKDRR B, in which the C-terminus is less than 10 residues from the final R domain. Most of the oomycete genomes surveyed encode two PKDRR members (one homologue each of subfamilies A and B), with that of *Aphanomyces astaci* containing an additional third potential PKDRR A subfamily homologue. Of these two subfamilies, PKDRR B members are most closely related to PKD proteins from metazoans, choanoflagellates and the nanoflagellate *Cafeteria roenbergensis* [27], which belongs to the same infrakingdom as oomycetes, the stramenopiles/heterokonts. The putative oomycete PKDRR channels are most closely related to the PKD2 and PKD2L subfamilies of metazoan PKD proteins. Of all the candidate eukaryotic channel proteins examined, only RyRs and oomycete PKDRRs contained R domains. In terms of their protein domain architectures, the simplest PKD proteins, such as the three homologues from *Cafeteria roenbergensis*, contain only a single PKD channel domain. Most oomycete PKDRR proteins contain a single PKD domain and two R domains, but some homologues possess additional, distinct domains N- or C-terminal to this predicted channel region, as shown in Figure 2B.

### 2.3. Analysis of the Predicted Channel Domains of Oomycete ITPR and PKDRR Channels

Key molecular features of ion channels are the conductance pathway, through which ions flow, and the selectivity filter, which allows some ions to enter while excluding others. These structural features are well defined in both the PKD2 and RyR/ITPR channel families. In metazoans, the selectivity filter of RyRs/ITPRs does not discriminate between cations particularly well, is located close to the C-terminus of these proteins and has the conserved structure of GGGVGD in ITPRs and of GGGIGD in RyRs. Both channel types are tetrameric, with the polar carbonyl groups of the glycine residues co-ordinating with cations within the three-dimensional structure of the filter [8]. A similar structure is likely to operate in oomycete ITPR channels, having the primary structure of either GGGIGD (metazoan RyR-like) or GGGLG(G/S) (found in some oomycete candidate ITPRs), as shown in Figure 3A. Leucine-containing selectivity filters have been identified in some of the ITPR homologues present in the alveolate *Paramecium tetraurelia* [28]. Further, mutation screening of mammalian ITPRs revealed that the valine of the selectivity filter could be substituted with an alanine residue without loss of ion permeation, indicating that the amino acid at this position is not critical for channel function [29].

In all ITPR/RyR proteins analysed in the current study, between 18 and 25 residues C-terminal of the selectivity filter lies a 25 residues pore-lining helix. This structure is conserved in these Ca^2+^-release channels and also shares limited identity with voltage-gated Ca^2+^ channel ion permeation pathways [8,28]. Even though the IT domain was not detected in all proteins using the CDD tool, all oomycete ITPR homologues characterised here share the conserved selectivity and pore-lining features, suggesting that they have the potential to operate as cation channels.

The candidate oomycete PKDRR proteins identified in this study contain a distinct selectivity filter and conductance pathway from those of the ITPR/RyR superfamily, that are related to those present in metazoan PKD channels. The predicted selectivity filters of oomycete PKDRR A channels (LGA) are similar to those found in mammalian PKD2-like channels (LGD) [30]. The PKDRR B channels possess a distinct candidate selectivity filter, of sequence AG(D/E). Immediately C-terminal of this is a conserved P-loop structure, that links to the S6 transmembrane, pore-lining helix of the channel protein. These observations suggest that the putative oomycete PKDRR proteins form a family of cation channels that are not closely related to the ITPR/RyR superfamily, despite sharing similar R domains with the later. Instead, these proteins are more closely related to the PKD family of the TRP superfamily of cation channels. However, these oomycete proteins are currently hypothetical, and to date there is no evidence that they are actually produced within these organisms. Consequently, we investigated if PKDRR proteins are present in *P. infestans*.

### 2.4. Biochemical Characterisation of the PKDRR B Protein from Phytophthora infestans

To investigate whether hypothetical PKDRR proteins are actually produced in oomycetes, three distinct custom rabbit polyclonal antibodies (pAb XP1, XP2 and XP3) were generated against *P. infestans* PKDRR B protein, using the synthetic peptides indicated in Figure 4A as immunogens. PKDRR B was selected as a target, rather than PKDRR A, because the former is more closely related to mammalian PKD2/PKD2L proteins, whose structural, electrophysiological and pharmacological properties are well characterised [31]. Use of these antisera in Western immunoblots to detect PKDRR proteins in lysates of *P. infestans* sporangia or zoospores demonstrated that pAb XP2 and pAb XP3 detected a major protein of 101 kDa apparent molecular weight, in both life-cycle stages. Antiserum pAb XP1 gave no strong signals on Western immunoblot. The protein detected by pAb XP3 was more abundant in sporangia than in zoospores, as shown in Figure 4B. It was also lower in molecular weight than predicted for *P. infestans* PKDRR B, which has a calculated weight of 159 kDa.

One possibility for the discrepancy between apparent and calculated molecular weight is that the PKDRR B protein could have undergone proteolytic cleavage. This is particularly pertinent for PKD channels, since mammalian PKD1 proteins are reported to undergo autocatalytic proteolysis at a motif termed the G-protein-coupled receptor proteolysis site (GPS) [32]. However, no GPS motifs were detectable in PKDRR channels, search using the SMART protein domain annotation resource [33]. Furthermore, the epitopes of pAb XP2 and pAb XP3 are located close to the C- and N-termini of *P. infestans*, making proteolysis an unlikely explanation for this anomaly. An alternative reason for this discrepancy could be anomalous migration of the PKDRR B protein on SDS-PAGE gels. This has been observed for mammalian RyRs: for example, full-length rabbit RyR1 has a calculated molecular weight of 565 kDa, but was originally reported to have an apparent molecular weight of 350 kDa estimated using SDS-PAGE [34]. However, our heterologous expression experiments indicate that this is unlikely, since recombinant proteins containing the PKDRR B protein migrate at similar apparent molecular weights to those calculated from their sequences, as shown in Figure 5A. Alternative splicing of the mRNA encoding PKDRR B could also explain its anomalous apparent molecular weight. However, the gene encoding PKDRR B is predicted to contain only two exons [35], one of which is too small (with the other encoding most of the protein) to account for the change in apparent molecular weight detected, Appendix A. Another possibility is the “protein splicing” of an intein from the PKDRR B protein. Several oomycete proteins are known to contain inteins, including RNA polymerase III from *P. ramorum* [36]. Inteins are defined by several consensus features, as reviewed in [37]. Of these, a cysteine residue (C), a poorly conserved endonuclease domain and a histidine-aspartic acid-serine (HDS) tripeptide define a candidate intein within *P. infestans* PKDRR B protein [38], that is of suitable size (57 kDa) to explain the discrepancy between the apparent and calculated molecular weight, Appendix A. It should be noted that many of the canonical features are absent from the putative PKDRR B intein, but neither are these essential for all experimentally verified examples of intein protein splicing [37]. Alternatively, pAb XP2 and pAb XP3 might be recognise proteins that are distinct from PKDRR, but contain sequences that are similar to those of the peptide immunogens. This is unlikely, because BLAST searching indicates that there are no closely related sequences encoded in the *P. infestans* genome within proteins of similar molecular weight to those detected on Western immunoblots by these antibodies, Appendix A.

To gain insights into the quaternary structure of *P. infestans* PKDRR B channel complexes, proteins from sporangial lysates were crosslinked with disuccinimidyl suberate (DSS), a bifunctional reagent that reacts with amine groups on proteins, forming a bond that is not hydrolysed by the conditions used in reducing SDS-PAGE. Analysis of crosslinked proteins by Western immunoblotting with pAb XP3 revealed major protein bands of 101 kDa and 326 kDa apparent molecular weight, as shown in Figure 4C. The apparent molecular weights of these protein bands are consistent with them being monomeric and crosslinked trimeric forms of the PKDRR B protein. Members of the PKD family have been reported to exist as dimeric, trimeric or tetrameric ion channel complexes [39]. The 326 kDa band is also detected in non-crosslinked samples, suggesting that this complex is resistant to disruption by reducing SDS-PAGE, as shown in Figure 4B,C.

Whereas ITPRs and RyRs are predominantly located in endomembrane systems such as the ER, PKD family members typically reside in the cell-surface membrane. To characterise the subcellular localisation of PKDRR B proteins, sporangia from *P. infestans* were immunostained with pAb XP3 and with a monoclonal antibody IID8, that recognises the type 2 SR/ER Ca^2+^ ATPase (SERCA2) pump, a marker of the ER. Sporangia were also stained with the fluorescent DNA chelator 4′,6-diamidino-2-phenylindole (DAPI), to identify nuclei arising from the process of zoosporogenesis. These experiments indicate that the PKDRR B protein is predominantly located in the cell-surface membrane of the sporangium, whereas SERCA2 is mainly found within the dividing zoospores, identified by positive DAPI-staining, as shown in Figure 4D. PKDRR could not be detected in any zoospore structures. This is consistent with PKDRR having a cell-surface localisation, a feature shared with most other members of the PKD channel family. Staining of sporangia with pAb XP3 was specific, since application of the pre-immune serum for this antibody, or pre-incubation with an excess of the synthetic peptide against which this antiserum was raised, gave no detectable signals, Appendix A. Similar immunostaining of sporangia was observed with the N-terminal directed antibody, pAb XP2. IIF microscopy of mycelia of *P. infestans* revealed that PKDRR was also located in the cell-surface membrane in this life-cycle stage, and was not detected in the lumen of the hyphal structures, where DAPI-positive nuclei were located, as shown in Figure 4E.

### 2.5. Generation of a Heterologous Expression System for Analysis of PKDRR B Channel Function

To determine whether *P. infestans* PKDRR B could function as a Ca^2+^ channel, we transiently transfected a human embryonic kidney cell-line (HEK-293T) with expression plasmids encoding Green Fluorescent Protein (GFP)- and FLAG-tagged forms of this protein. SDS-PAGE and Western immunoblotting with anti-GFP or anti-FLAG antibodies revealed that transfection was successful: within 24 h of transfection, HEK-293T cells expressed proteins matching their calculated molecular weights (188 kDa for GFP-PKDRR B and 160 kDa for FLAG-PKDRR B), as shown in Figure 5A. Microscopic imaging of GFP fluorescence was used to identify individual cells successfully transfected with GFP alone, or with the GFP-tagged PKDRR B construct. GFP fluorescence was uniformly distributed in cells expressing GFP alone but was detected in the membranes of cells producing GFP-PKDRR B, as shown in Figure 5B, supporting a cell-surface localisation of this protein. To detect Ca^2+^ responses to known PKD agonists, transfected HEK-293T cells were loaded with the ratiometric Ca^2+^-sensing dye fura-2 and imaged by fluorescent videomicroscopy. The resting fura-2 ratio was indistinguishable between GFP- and GFP-PKDRR-transfected cells (values of 0.493 ± 0.078 and 0.439 ± 0.089, respectively; mean ± standard error of 6 independent experiments and 35 or 42 cells; not significant by unpaired Student’s T-test). Neither GFP- nor GFP-PKDRR-transfected cells displayed reproducible increases in fura-2 ratio (corresponding to rises in cytoplasmic free [Ca^2+^]) in response to the PKD agonist triptolide [40], nor to cinnalmaldehyde, a compound that increases Ca^2+^ levels in the zoospores of *P. capsici*, through a mechanism proposed to operate through activation of members of the TRP channel superfamily [41]. Both GFP and GFP-PKDRR expressing cells displayed robust and reproducible responses to ATP, a purinoceptor ligand that activates Ca^2+^ release from the ER by stimulation of phospholipase C and production of IP_3_ [42], as shown in Figure 5C. These measurements indicate either that the agonists used did not activate the heterologously expressed PKDRR B channels, that PKDRR proteins are not cation channels, or that they are cation channels, but they are not produced in a functional state in the HEK-293T system. We consider that the latter possibility is the most likely, since difficulties in expressing functional ion channels from non-mammalian organisms have been reported. For example, expression of functional voltage-gated Ca^2+^ channels from *Trichoplax adhaerens* (a basal animal) was only possible when HEK-293T cells were co-transfected with an additional, non-channel forming subunit of this channel complex [43]. In future, genetic engineering approaches such as Crispr-Cas9 mediated knockout of genes in *Phytophthora* [44], will help to elucidate the roles of the PKDRR protein family in oomycete physiology.

## 3. Conclusions

The current study demonstrated that candidate PKDRR channels are taxonomically restricted to oomycetes. They are distinct from the ITPR/RyR superfamily of Ca^2+^-release channels in terms of their phylogeny, predicted channel and domain structures, and subcellular localisation. Instead, these putative oomycete cation channels belong to the polycystin/PKD protein family, which is part of the TRP superfamily. Features that place oomycete PKDRR proteins in this family are: (i) phylogeny, in that their closest relative is a PKD protein from the stramenopile/heterokont *Cafeteria roenbergensis*; (ii) domain architectures, which most closely resemble those of other PKD family members. They only share RyR domains (R) with the RyR family and show no other detectable homology with the ITPR/RyR superfamily; (iii) their putative channel domains are similar to those of other PKD family members and are distinct from those of the ITPR/RyR superfamily; and (iv) they are predominantly located in the cell-surface membranes of *P. infestans* mycelia and sporangia, in contrast to typical ITPR/RyR superfamily proteins, which mainly reside in endomembrane systems. We also provide experimental evidence that the hypothetical PKDRR B protein is actually produced in *P. infestans* and forms high-molecular-weight, multimeric complexes. We were not able to reconstitute the function of these candidate PKDRR B channels in a heterologous expression system. However, given that two well-conserved PKDRR homologues were present in most oomycete genomes analysed, and that these proteins appear to be absent from other taxa, they make attractive targets for the development of novel anti-oomycete “fungicides”. Understanding the physiological roles of PKDRR proteins in oomycetes will be an essential step in this process.

## 4. Materials and Methods

### 4.1. Materials

Three distinct rabbit polyclonal antibodies (XP1, XP2, XP3) were raised against synthetic peptides derived from *P. infestans* PKDRR B protein (Accession Number: XP_002908895.1), using the PolyExpress Gold Package service (GenScript, Hong Kong, China). The sequences of these immunogenic peptides are shown in Figure 4A. They were selected based on their predicted antigenicity and limited identity with other oomycete or human proteins, Appendix A. Antibodies were isolated from the resulting antisera by the manufacturer, using affinity purification against the cognate immunogenic peptide. BioRad SDS-PAGE apparatus, Western Transfer units and Precision Plus prestained molecular weight protein markers were from Brennan & Company (Stillorgan, Co. Dublin, Ireland). Disuccinimidyl suberate (DSS), Lipofectamine LTX, OptiMeM transfection media, Molecular Probes fura-2-acetoxymethyl ester (fura-2-AM), Amersham Protran Nitrocellulose Membranes and ECL Plus Western Blotting Substrates and mouse anti-GFP monoclonal antibody clone JL-8 were from Fisher Scientific Ireland (Ballycoolen, Dublin, Ireland). Anti-SERCA2 monoclonal antibody IID8 was from Santa Cruz Biotechnology (CA, USA). Horse-radish peroxidase-conjugated secondary antibodies, monoclonal antibody M2 recognising the FLAG tag epitope, cinnamaldehyde, ATP, RIPA buffer, protease inhibitor mixture I and paraformaldehyde were from Sigma-Aldrich (Arklow, Co. Wicklow, Ireland). Cy3-conjugated anti-rabbit and Cy5-conjugated anti-mouse secondary antibodies were from Jackson ImmunoResearch Europe Ltd. (Ely, Cambridgeshire, UK). Triptolide was from Enzo Life Sciences (Exeter, UK). All other reagents were of analytic grade or better and were obtained from Sigma-Aldrich, Ireland.

### 4.2. Evolutionary History of Phytophthora Infestans ITPR and PKDRR Proteins

*Homo sapiens* ITPR1 (Accession Number: NP_001093422.2) was selected as the starting point for investigating the evolutionary history of oomycete ITPR proteins, as it is a well-characterised member of this ion channel family. Oomycete homologues of *H. sapiens* ITPR1 were identified using a modified protein basic alignment of sequences tool at NCBI’s Pubmed website [45], essentially as described previously [10,12]. In this search, eight representative metazoan ITPR homologues were recorded, then metazoa were excluded from subsequent searches. This led to retrieval of 61 additional proteins sequences from non-metazoan eukaryotes, of which, 47 were oomycete ITPR homologues. Details of these proteins, including their accession numbers, length in amino acid residues, percentage identity and coverage (“overlap”) with *H. sapiens* ITPR1, and E-value (the probability that the homologues identified are related to *H. sapiens* ITPR1 by chance, taking 1 × 10^−5^ as the threshold), are recorded in Appendix A. The domain architectures of these proteins were investigated using the conserved domain database (CDD) at PubMed [24] and are also indicated in this Table. The evolutionary relationships of these proteins were investigated using the maximum likelihood method following MUSCLE multiple sequence alignment (MSA) [46], using default settings on MEGA-X software [21]. This included a bootstrap analysis of the statistical validity of clustering of groups of proteins, taking values above 50 as significant. For presentation of selectivity filter and pore sequence alignments, MSA of selected protein channel domains was performed using the CLUSTAL Omega programme [47].

Essentially similar approaches were performed to reconstruct the evolutionary history of oomycete PKDRR channels. In this case, the starting sequence was PKDRR A protein from *P. infestans* (Accession Number: XP_002909214.1). The results of these analyses are summarised in Appendix A.

To investigate reasons for the discrepancy between the calculated molecular weight of the *P. infestans* PKDRR B protein and that estimated on SDS-PAGE gels, we investigated the intron-exon structure of the gene encoding it, using the IntronDB resource [35]. This was used to examine the possibility that alternate splicing of mRNA could explain the discrepancy. To explore the possibility that the *P. infestans* PKDRR B protein could contain an intein, the InBase resource [38] was utilised at the website: http://www.biocenter.helsinki.fi/bi/iwai/InBase/tools.neb.com/inbase/identify.html/.

### 4.3. Culture of Phytophthora Infestans

All procedures were performed using sterile technique and cultures were grown on 10 cm Petri dishes. *P. infestans* strain 88069 (mating type A1) was obtained The Sainsbury Laboratory (Norwich, UK) and was cultured on Rye A agar in the dark at 20 °C, as described by Caten and Jinks [48]. To induce formation of sporangia, a 6 mm plug was cut from the edge of a 7-day-old culture and transferred to Rye B agar. Cultures were maintained for an additional 7 days, or until sporangia were observed using brightfield microscopy. Sporangia were extracted in 5 mL of modified Petri’s solution (MPS, 5 mM CaCl_2_, 1 mM MgSO_4_, 1 mM KH_2_PO_4_ and 0.8 mM KCl) by scraping the surface of the culture with a glass rod. Sporangia were transferred to a 50 mL tube and the plates were washed with another 5 mL of MPS. The two sporangial suspensions were pooled and filtered through a 50 μm nylon mesh into a fresh 50 mL tube to remove fragments of mycelium and agar. To induce the release of zoospores, sporangia were transferred to a 4 °C cold room, in the dark, for 3 h. Subsequently, zoospores were separated from sporangia by filtration through a 10 μm nylon mesh [49]. To extract protein lysates from sporangia or zoospores, these were collected by centrifugation at 3000 g for 15 min at 4 °C. Pellets were resuspended in 5 volumes of extraction buffer (20 mM Tris pH 7.5, 5% glycerol, 1% SDS, 10 mM DTT and protease inhibitors) by twenty passes of a glass-teflon homogenizer at 4 °C. To prepare samples from mycelia, these were frozen in liquid nitrogen, ground to a powder using a pestle-and-mortar, then were resuspended in 5 volumes of extraction buffer. All lysates were centrifuged at 20,000× *g* for 60 min at 4 °C, then the resulting supernatants frozen in liquid nitrogen and stored at −80 °C until use.

### 4.4. Heterologous Expression of the P. infestans PKDRR B Protein in Human Embryonic Kidney Cell-line

Mammalian codon-optimised DNA segments (locus_tag = “PITG_00274” and “PITG_00635”) of *P. infestans* T30-4 gene ryanodine-Iinositol 1, 4, 5-triphosphate receptor Ca^2+^ channels (GenBank: DS028118.1) were engineered with a FLAG epitope tag (nucleotide: ATG GAC TAC AAGGAC GAC GAC GAC AAA; peptide: MDYKDDDDK) at the 5′ end of each, then were subcloned into the *KpnI*/*BamHI* site of the cloning vector pUC57 (GenScript Inc., Piscataway, USA). The resulting FLAG-tagged constructs (FLAG-274, 4299 bp; FLAG-635, 3918 bp) were confirmed by restriction enzyme digestion, then were subcloned into the *KpnI*/*BamHI* site of *Kan^R^* pAcGFP1-C1 vector (pAcGFP1-C1 was a gift from Michael Davidson (Addgene plasmid # 54607)). The resulting plasmids were propagated in *E. coli* (strain DH5α) using LB medium containing 50 µg/mL kanamycin and were purified using a Qiagen EndoFree Plasmid Maxi Kit (Qiagen, Manchester, UK), according to the manufacturer’s instructions. The sequences of these constructs were checked and confirmed by using the commercial TubeSeq Service provided by Eurofins Genomics (Eurofins MWG Operon Inc., Ebersberg, Germany).

Human embryonic kidney-293T (HEK-293T) cells, from LGC Standards (Middlesex, UK), were cultured in Dulbecco’s Modified Eagles Medium containing 10% foetal bovine serum, 100 units/mL penicillin and 100 μg/mL streptomycin, at 37 °C in humidified 5% CO_2_/95% air, according to the supplier’s instructions. For transient transfections, cells were either subcultured on 6-well plates (Sarstedt Ireland, Co. Wexford) or 35 mm glass-bottom bottom dishes, with 10 mm in diameter, 0 thickness coverslips (MatTek Life Sciences, Ashland, MA, USA). Cells were transfected using Lipofectamine LTX according to the manufacturer’s instructions, with a DNA concentration of 1 μg/mL found to be optimal for each well of a 6-well plate. After 24 h, cells transfected with either GFP (empty pAcGFP vector) or with GFP-PKDRR were visualised using a fluorescent videomicroscopy system, essentially as described previously [50]. For Ca^2+^-imaging experiments, transfected cells that were grown on glass-bottom dishes were washed twice with 1 mL of modified Krebs-Henseleit Buffer (KHB: 120 mM NaCl, 4.8 mM KCl, 2 mM CaCl_2_, 1.2 mM MgSO_4_, 1.2 mM KH_2_PO_4_, 25 mM NaHCO_3_, 5 mM HEPES), then were incubated with 2 μM of fura-2-AM in 250 μL of KHB for 30 min at 37 °C. This esterified dye can cross cell membranes, but the AM group is removed by cellular esterases, trapping the Ca^2+^-sensing free-acid form in the cytoplasm. Cells were then washed twice with 1 mL of KHB, then were maintained in 980 μL of this medium. Loaded dishes were transferred to the stage of an Olympus IX51 inverted microscope maintained at 37 °C within an encapsulating incubator. GFP-positive cells were identified by excitation at a wavelength of 490 nm via an Olympus UplanF1 1.3 NA 100× oil-immersion objective, selected from the light produced by a 75 W Xenon lamp using a Cairn Monochromator (Faversham, Kent, UK), and collecting emitted light through a 530 nm cut-off dichroic mirror. Images were captured at 2 Hz using an ORCA ER CCD videocamera (Hamamatsu Photonics Ltd., Hertfordshire, UK) and Andor IQ v1.9 acquisition software (Belfast, Northern Ireland). For Ca^2+^-imaging of GFP positive cells, fura-2 was excited alternately at wavelengths of 340 nm (Ca^2+^-bound form of fura-2) and 380 nm (Ca^2+^-free form of the dye), with an exposure time of 500 ms/frame. Emitted light was filtered through a 400 nm bandpass dichroic mirror. Following recording of resting values for 50 s, agonists (1 mM ATP, 100 nM triptolide, or 10 mM cinnamaldehyde) were added as 20 μL bolus additions of 50 x concentrated stocks. Images were recorded for another 200 s. Captured images were masked to remove background, then were processed to generate ratio values (340/380 nm), which are proportional to the cytoplasmic free Ca^2+^ concentration [51]. For each GFP-positive cell imaged, fura-2 ratio values were plotted against time using Microsoft Excel software.

### 4.5. Protein Characterisation

Protein concentrations were determined using the method of Bradford [52], with bovine serum albumin as a standard. For crosslinking experiments, 50 μg of protein extracted from *P. infestans* sporangia was incubated with either 100 μM of DSS or a solvent control (1% dimethyl sulfoxide). After 30 min incubation at 20 °C in darkness, reducing SDS-PAGE sample buffer was added and proteins were stored at −80 °C prior to further analyses. SDS-PAGE and Western immunoblotting were performed essentially as described previously [53]. For most purposes, 7.5% resolving gels were used, but 5% SDS-PAGE was used to resolve the large, multimeric protein complexes generated in crosslinking experiments. Affinity-purified rabbit polyclonal antisera against synthetic peptides from *P. infestans* PKDRR B protein (pAbs XP1, XP2 and XP3) were used at a dilution of 1:100; JL-8 mouse anti-GFP at 1:1000; and anti-FLAG mouse monoclonal antibody at a concentration 1 μg/mL Both anti-rabbit and anti-mouse horse-radish peroxidase conjugates were used at dilutions of 1:2000. Prestained molecular weight markers were used to estimate apparent molecular weights of *P. infestans* proteins, estimated from the linear equation fit of retention factors (= distance a protein has migrated/total length of the gel) against the known molecular weights of standards. In some cases, Western immunoblots were developed using a LiCor Odyssey near-infrared scanner (LI-COR Biosciences UK Ltd., Cambridge, UK). In this case, a 1:15,000 dilution of IRDye^®^ 680RD donkey anti-rabbit IgG (H + L) was used to detect the primary antibodies.

### 4.6. Indirect Immunofluorescent Microscopy

To determine the subcellular location(s) of PKDRR B protein in *P. infestans*, we employed confocal indirect immunofluorescent microscopy, essentially as described by Ah-Fong and Judelson [54]. Sporangia or mycelia were collected as described in Section 4.2, then were fixed for 30 min at 20 °C in 1 mL of 4% paraformaldehyde 50 mM piperazine-N,N′-bis(2-ethanesulfonic acid) (PIPES) pH 6.8. Tissues were collected by centrifugation at 1000 g for 5 min, washed in 1 mL of 100 mM PIPES pH 6.8, centrifuged, then were washed again in 1 mL of phosphate buffered saline (PBS). Tissues were permeabilised using 1 mL of 0.2% Triton X-100 in 100 mM PIPES pH.6.8 for 30 min at 20 °C, then were washed twice with 1 mL of PBS (by centrifugation and resuspension). Non-specific binding was blocked by incubation in 1% BSA/0.1% gelatine in PBS for 1 h. Tissues (15 μL/experiment) were then incubated with the following dilutions of primary antibodies in 1% BSA/0.1% gelatine/PBS for 1 h: 1:50 pAb XP3, or XP3 pre-immune immunoglobulins, plus 1:100 anti-SERCA2 (clone IID8). In some experiments, pAb XP3 was incubated with 10 μg/mL of its cognate immunogenic peptide for 1 h prior to addition, as a negative control. Next, tissues were washed twice with 1 mL of PBS, then were incubated for 40 min with 1:200 Cy3-conjugated anti-rabbit secondary, 1:200 Cy5-conjugated anti-mouse secondary antibody and 1 ng/mL DAPI. Tissues were washed twice with 1 mL of PBS, then were mounted in Mowoil medium on a 10 mm coverslip, placed on a microscope slide. Once the samples had dried, tissues were imaged using an Olympus FV100 Confocal microscopy system, at the Bioscience Imaging Centre, Department of Anatomy and Neuroscience, University College Cork.

## Figures and Tables

**Figure 1 pathogens-09-00577-f001:**
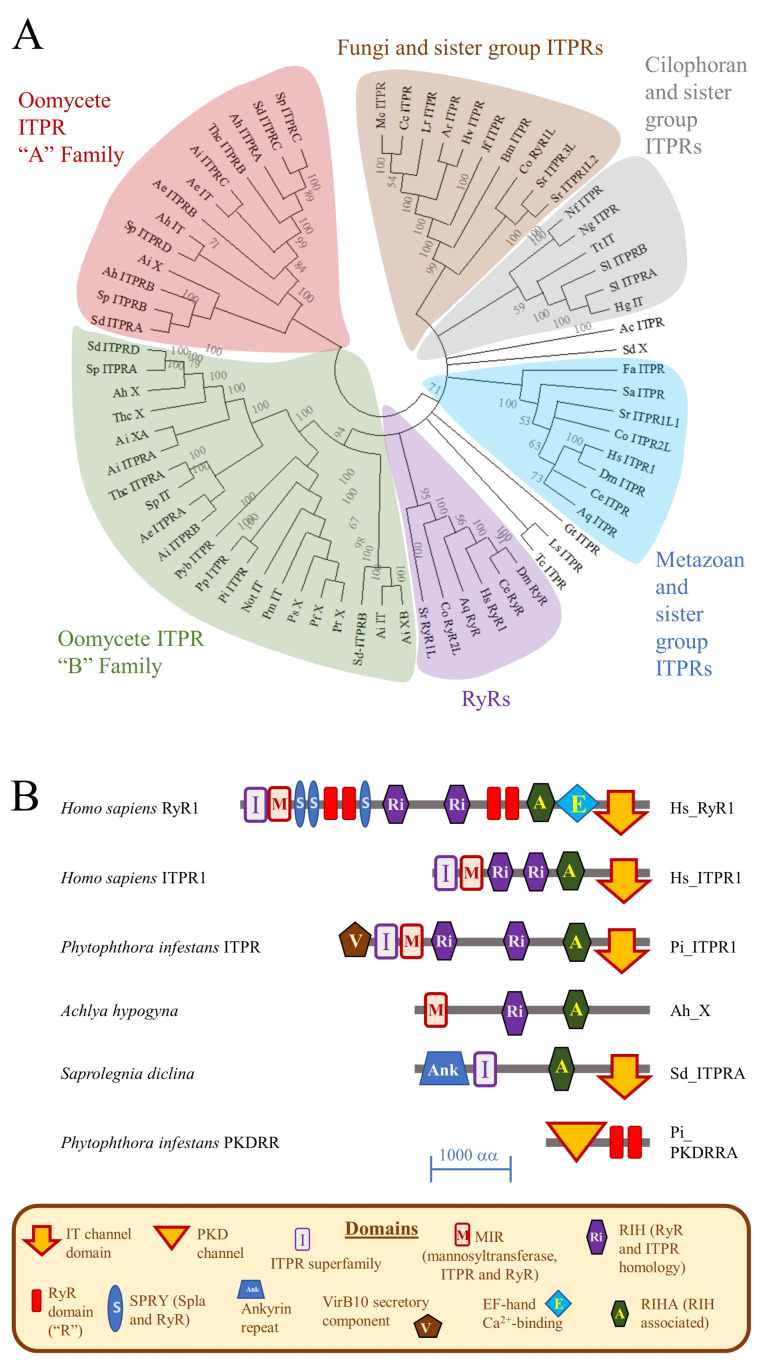
Reconstruction of the Evolutionary History and Protein Domain Architectures of Oomycete ITPR homologues. (**A**) The evolutionary history of oomycete ITPR homologues was reconstructed using the maximum likelihood method on MEGA-X software. The numbers at nodes represent the results from a bootstrap analysis (500 replicates) of the groupings of proteins used to statistically determine the validity of this clustering. Bootstrap values of less than 50 are considered insignificant and are not shown. The suffix following each sequence identifier refers to predictions of protein domain architectures: “ITPR” means ITPR-like; “IT” indicates that an ion channel domain was identified, but no other ITPR-like features; “X” signifies that no IT domain was detected, even though other ITPR-like domains were found. Where multiple homologues were found, they are followed by a letter in alphabetical order: this has no significance, other than the order in which they were ranked in the BLAST search. Key organism codes are: Hs, *Homo sapiens*; Sr, *Salpingoeca rosetta*; Pi, *Phytophthora infestans*; Pp, *P. palmivora*; Sd, *Saprolegnia diclina*; Sp, *S. parasitica*; Ai, *Aphanomyces invadans*; Ae, *A. euteiches*; and Ah, *Achyla hypogyna*. (**B**) Domain structures of ITPR homologues were determined using the CDD tool. Representative examples are diagrammatically illustrated to scale (scale bar indicates 1000 amino acid residues). Full details of protein accession numbers, identity shared with *H. sapiens* ITPR1 and protein domain architectures are given in Appendix A.

**Figure 2 pathogens-09-00577-f002:**
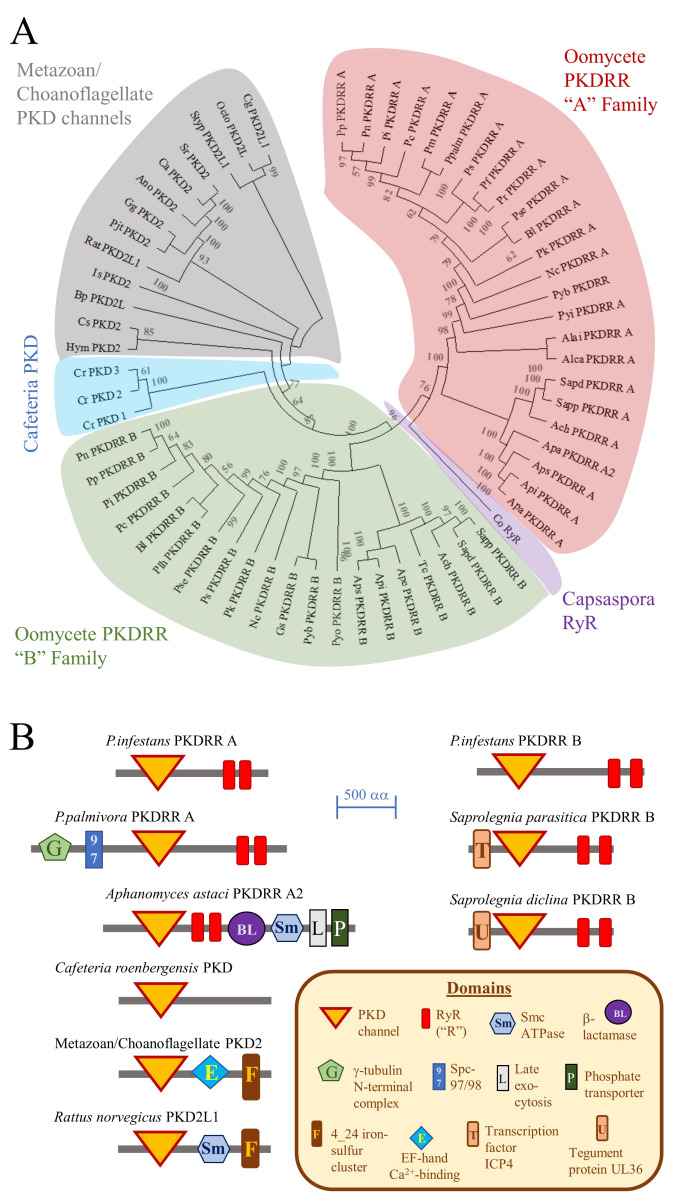
Reconstruction of the Evolutionary History and Protein Domain Architectures of Oomycete PKDRR proteins. (**A**) The evolutionary history of oomycete PKDRR homologues was reconstructed using the maximum likelihood algorithm on MEGA-X software. The numbers at nodes are from a bootstrap analysis (500 replicates) of the clustering of proteins, with values of less than 50 being considered insignificant. Key organism codes are: Rat, *Rattus norvegicus*; Sr, *Salpingoeca rosetta*; Cr, *Cafeteria roenbergensis*; Pi, *Phytophthora infestans*; Pp, *P. parasitica*; Ppalm, *P. palmivora*; Sapd, *Saprolegnia diclina*; Sapp, *S. parasitica*; Api, *Aphanomyces invadans*; Ape, *A. euteiches*; and Apa, *A. astaci*. (**B**) Domain structures of ITPR homologues were determined using the CDD tool at NCBI. Representative examples are diagrammatically illustrated, to scale (scale bar indicates 500 amino acid residues). Full details of protein accession numbers, identity shared with *P. infestans* PKDRR A and protein domain architectures are given in Appendix A.

**Figure 3 pathogens-09-00577-f003:**
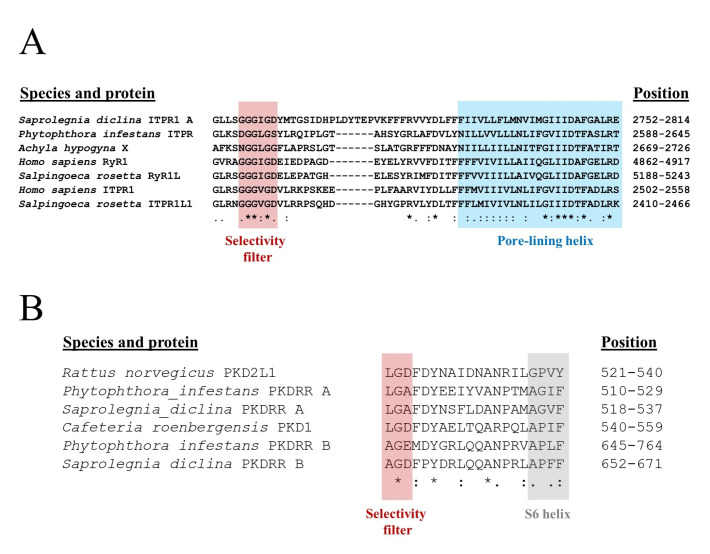
The Predicted Selectivity Filters and Channel Pores of Oomycete ITPR and PKD Homologues. (**A**) Multiple sequence alignment (MSA) of the selectivity filter and pore region of ITPRs and RyRs from a mammal (*H. sapiens*), a choanoflagellate (*Salpingoeca rosetta*) and two oomycete species (*P. infestans* and *S. diclina*). (**B**) MSA of the selectivity filter and permeation pathway of PKD channels from a mammal (*Rattus norvegicus*), a nanoflagellate (*Cafeteria roenbergensis*) and two oomycetes (*P. infestans* and *S. diclina*). Numbers to the right of each MSA indicate the position of the sequence in number of amino acid residues. Under each MSA, **“*”** indicates complete conservation of that residue, **“:”** shows homology between amino acids at that site; and **“.”** indicates that the residues at that site belong to the same group (e.g., small, hydrophobic, or charged amino acids).

**Figure 4 pathogens-09-00577-f004:**
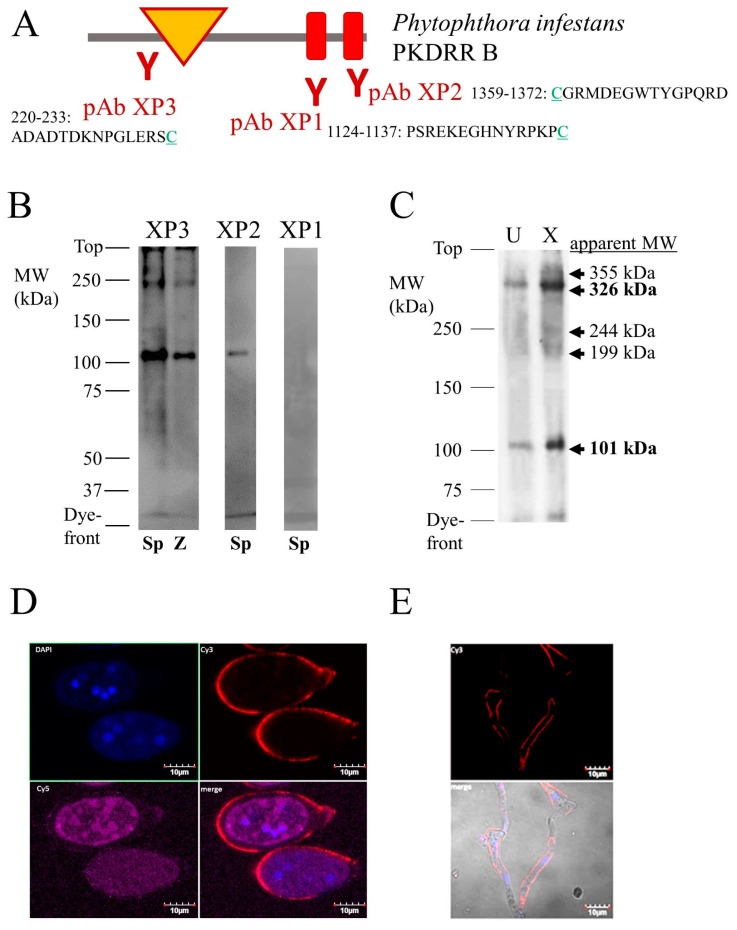
Characterisation of the PKDRR B Protein in *P. infestans.* Panel (**A**) shows a schematic of the *P. infestans* PKDRR B protein, indicating the sites against which three anti-peptide antisera (pAb XP1, XP2 and XP3) were generated. The cysteine residue (C) at either the N- or C-terminus of each peptide is not endogenous, but was added to facilitate coupling to a carrier protein, as part of the immunisation regime. In panel (**B**), 50 μg of protein from lysates of either sporangia (Sp.) or zoospores (z) from *P. infestans* was resolved on 7.5% SDS-PAGE gels, transferred on to nitrocellulose and immunostained with the antisera against PKDRR B. Numbers to the left of this image indicate the positions of molecular weight markers. (**C**) shows 50 μg of *P. infestans* sporangial lysate protein, incubated in the presence (X) or absence (U) of the crosslinker DSS, then resolved by 5% SDS-PAGE, transferred on to nitrocellulose, then immunostained with pAb XP3. Numbers to the left of this image indicate the positions of molecular weight markers; numbers to the right the apparent molecular weight of proteins or protein complexes detected by this antiserum. Panel (**D**) shows indirect immunofluorescent micrographs of *P. infestans* sporangia stained with the DNA-binding fluorescent dye DAPI (“DAPI”), antiserum pAb XP3 against PKDRR B (“Cy3”) or a monoclonal antibody IID8, recognising the SR/ER-Ca^2+^-ATPase pump (SERCA2, “Cy5”). Panel (**E**) is of a similar IIF microscopy experiment, examining the distribution of PKDRR B and DAPI staining in mycelia. The “merge” image also shows a brightfield micrograph of these hyphae.

**Figure 5 pathogens-09-00577-f005:**
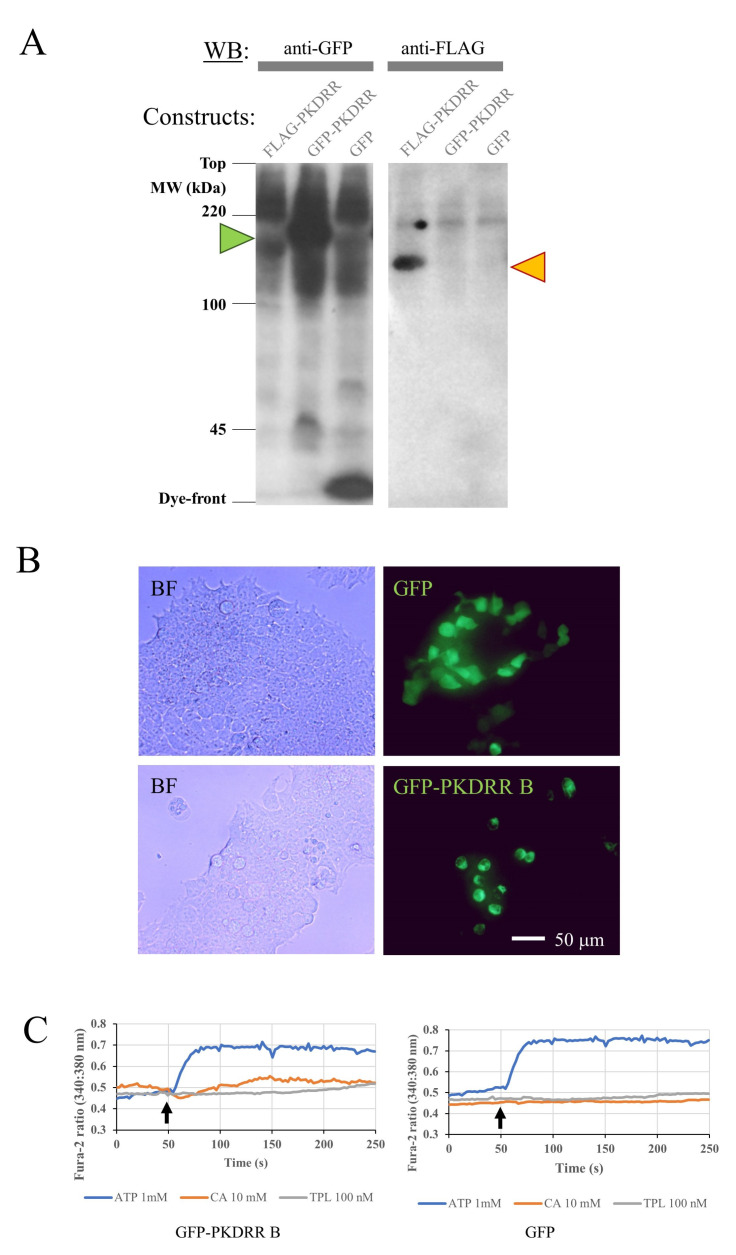
Heterologous Expression of *P. infestans* PKDRR B Protein in HEK-293T Cells. Mammalian HEK-293T cells were transiently transfected with plasmids encoding GFP alone (GFP), or with FLAG- (FLAG-PKDRR) or GFP-tagged (GFP-PKDRR) forms of the *P. infestans* PKDRR B protein. (**A**) After 24 h, equal quantities of protein (30 μg protein/lane) from lysates of these transfected cells were resolved on 7.5% SDS-PAGE gels, transferred to nitrocellulose membranes by Western blotting and tagged proteins detected using antibodies recognising GFP or the FLAG epitope. The green triangle indicates the position of GFP-PKDRR B (188 kDa); the orange triangle that of FLAG-PKDRR B (160 kDa). (**B**) HEK-293T cells were transfected with GFP protein alone, or with GFP-tagged PKDRR B and were imaged using brightfield (BF) or fluorescent (GFP) microscopy. (**C**) HEK293T heterologously expressing either GFP alone (GFP) or GFP-tagged PKDRR protein (GFP-PKDRR) were loaded with the Ca^2+^-sensing dye fura-2. Following recording of a 50 s baseline, responses to addition (arrows) of either 100 nM triptolide (TPL), 10 mM cinamaldehyde (CA), or 1 mM ATP were recorded for another 200 s.

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
