# Peer review of "Polycystic Kidney Disease Ryanodine Receptor Domain (PKDRR) Proteins in Oomycetes"

_pathogens, 2020, doi:10.3390/pathogens9070577_

Round 1

Reviewer 1 Report

I like the idea, designed experiments and achieved results, however I would like to point out some minor remarks, which I believe can improve the way of result presentation.

While writing abbreviations of Latin names e.g. P.capsici you don’t use space after dot. Is this a pattern of the Journal? I don’t think so, it looks much better P. capsici. It occurs very often: L40, 7375, 81, 87, 96, 119, 145, 147, 164, 205, 244, 257, 262, 266, 276, 288, 300, 374, 376, 413, 424, 503, 510, 530, 531.

There are many extra spaces like in L48 before Extracellular Why? It is the same in L54, 57, 66, 82, 84, 88, 103, 106, 144, 149, 161, 166, 184, 196, 207, 214, 228, 264, 290, 294, 295, 296, 299, 389, 395, 423, 475, 484, 499, 515, 518, 520.

L69- misspelling of Pythium

L75- misspelling of P. cinnamomi

The graph above line 152 is poor quality and description below not well visible in B Latin names should be written with larger fonts

L 189 The dot is not needed after “A homologe” and “Our” should start with small “o”

Same problem on page L245 in the above graph, especially in A description could be hardly read

L350 – above 2 graphs are poor quality especially descriptions on axis are not readably

L514 and 515 after pH I suggest putting dot before 6.8

In references Latin names are not in italic, why? L-549, 568, 570, 573, 576, 578, 580, 581, 584, 599, 620, 626, 627, 636, 638, 639.

Reviewer 2 Report

The manuscript presents a series of arguments/parts that are not well connected; in addition, there are too many acronyms, often referred to as synonyms or joined together to add confusion to the extent that the manuscript is hardly readable. Some examples: line 141-142 " ITPR and RyR (MIR) domain; an RyR and ITPR Homology (RIH) domain; a RIH Associated (RIHA) domain; and an ion transport/channel (IT) domain”, and in the legend of Figure 1 “IT indicates an ion channel domain was identified, but no other ITPR-like features”; so is IT part of ITPR?

So, it is necessary a table summarizing all the acronyms.

Other points:

  • Paragraph 2.1: a) the topic was partially discussed previously by the Authors (ref. 9,10); b) Figure 1B shows several domains but the “IT” domain mentioned at line 142 is missing; c) the Phytophthora infestans PKDRR shows completely different domains so that it is difficult to understand why it is included there; concluding, the whole paragraph is of little interest in a study focused on the PKDRR channel family;
  • Paragraph 2.2: that is ok but Authors must specify the correlation between RyR and R domain (Figure 2B);
  • Paragraph 2.3: because the study is focused on the PKDRR channel family the part concerning ITPR channels could be omitted;
  • Figure 4 (Paragraph 2.4): Antibodies raised against peptides are certainly able to recognize peptides, but the recognition of a protein is more complex because more proteins can randomly contain the same amino acid sequences. Only if the three antibodies raised against peptides recognize an identical protein of the predicted MW there is confirmation (although not yet certainty) of the effectiveness and validity of antibodies as a tool for protein characterization; in this case I see a problem because the predicted MW is 159 kDa (line 266) but it was detected a subunit of 101 kDa (line ; moreover, because the two antibodies employed for Fig4b were raised against amino acid sequences starting from aa 220 and ending aa 1373 along the sequence of the infestans PKDRRB aa sequence, the polypeptide recognized simultaneously by pAbXP3 and pAbXP2 should have a molecular weight around 130 kDa and not 101 kDa; Authors must explain this contradiction.
  • Figure 4: Regarding pAbXP2 antibodies, the amino acids between position 1359 and 1373 (extremes included) are 15 but the sequence included in Figure 4A is only of 13 amino acids;
  • Finally, the PolyExpress Gold Package service (GenScript, Hong Kong) include an antigen affinity purification of the pAbs, but at line 499-500 Authors claimed an affinity purification against infestans PKDRRB protein: please explain if the protein was purified and eventually how, and how the affinity purification was performed.

For the above-mentioned items the MS needs a major revision

Round 2

Reviewer 2 Report

I repeat the previous judgment (major revision) for the following reasons:

MS clarity and acronyms

On the one hand I do not understand why the authors did not agree to add a table / list of acronyms, on the other hand it is not acceptable to answer with "we followed the instructions" when this was not done for references.

- Paragraph 2.1

I cannot see in the revised text an explanation regarding the fact that “In many oomycete proteomes, PKDRR proteins have been annotated as ryanodine receptor/inositol trisphosphate receptor calcium channels (or “RIR-CaC”)” as stated in the Authors’ response.

-Paragraph 2.2 - the correlation between RyR and R domain (Figure 2B)

I'm obviously not smart enough or a very simple mind, but if RyR and R are synonymous, why not use one term or keep them together? (line 177-179: “One of these is termed the RyR (R) domain and this was only detected in eukaryotic RyRs (most of which have 4 of these domains); in several families of oomycete proteins; and in multiple bacterial, archaeal and viral proteins [10]. The function of the R domain is unknown.”)

-Personally, I still think that because the study is focused on the PKDRR channel family the part concerning ITPR channels could be omitted.

-Figure 4 (Paragraph 2.4) and Antibodies

If GenScript have raised three different antibodies and Authors could show that only two of them was a success adding a western blot for the XP1 antibodies; also, the fact that the pAb XP2 and pAb XP3 antibodies selectively recognize a P. infestans protein is a good evidence that they effectively recognize the PKDRR B protein; BLASTs do not add evidence concerning the protein identification.

-In the revised MS at lines 282-285 it is indicated that “the epitopes of pAb XP2 and pAb XP3 are located close to the C- and N-termini of P. infestans, making proteolysis an unlikely explanation for this anomaly. Alternative reasons for this discrepancy in molecular weight might be alternate splicing of mRNA transcripts, or extensive post-translational modification of the protein, such as hyperphosphorylation”

Because the full-length PKDRR B protein is of 159 kDa and two antibodies recognize a polypeptide of approximately 101 kDa it means that 1/3 of the polypeptide / protein was lost; but because the distance in amino acids between the two epitopes correspond approximately to 130 kDa it is unlikely that a proteolysis from the N or C term of the polypeptide could have reduced the total length/weight by a third because one or other of the epitopes (XP2 or XP3) would have been eliminated not being far enough away from the extremes of the polypeptide. Therefore, I believe that the only reasons for this discrepancy could be an alternate splicing (and a search for possible sites of splicing in the DNA sequence cost nothing).

- Figure 4: Regarding pAbXP2 antibodies, the amino acids between position 1359 and 1373 (extremes included) are 15 but the sequence included in Figure 4A is only of 13 amino acids

Maybe I was not clear enough, but if a cysteine has been added to the amino acid sequences indicated for the peptides of Figure 4A, this must be indicated in the legend of this figure and in the M&M not simply in a supplementary Figure 1A. Anyway, adding a cysteine the length of the peptide increases not decreases; so, 15 amino acids are present between the position 1359 and 1373 of the protein sequence, but the XP2 peptide is of 13 amino acids in Figure4A: where are the other two amino acids and cysteine too? Please note also that in Supplementary Figure 1A the cysteine is included, but the length indicated for all the three peptides is 14, with the XP2 peptide (cysteine included) is indicated with 15 characters (CPSREKEGHNYRPKP) that differs from the indication in Figure 1A (GRMDEGWTYGPQR); moreover, the peptide PSREKEGHNYRPKP represents XP1 in Figure 4A and XP2 in the Supplementary Figure 1A, whereas XP2 of Figure 4A (GRMDEGWTYGPQR) is not included in Supplementary Figure 1A. This confusion leads to the question of what peptides were actually used to produce antibodies.

Round 3

Reviewer 2 Report

This paper remains the problem of the size of the protein recognized by antibodies. Anyway, the manuscript is clearly improved.

But few details still need to be corrected:

  1. Figure 4 A: pAb XP2 start end should be 1359-1372 (and not 1359-1371) otherwise an amino acid is missing / the sum is not 15;
  2. Figure 4 A and Figure S1 R2 B show different things. In fact, from Figure 4 the order is pAb XP3 (start 220) - pAb XP1 (start 1129) - pAb XP2 (start 1359), whereas in Figure S1 R2 B the order is pAb XP2 (start 220) - pAb XP1 (start 1129) - pAb XP3 (start 1359); it is necessary to eliminate discrepancies;
  3. Moreover, in Figure S2 R2 A the order correspond to Figure 4 (please eventually correct so that all figures show identical details) but, very important, the amino acid sequence for XP1 (NTSSGRERRESGGPC) is not that indicated in Figure S2 R2 A (PSREKEGHNYRPKP) neither that included in Figure 4 A (PSREKEGHNYRPKP), and also is not the sequence of Figure S2 R2 B (PSREKEGHNYRPKP).

The authors should verify which synthetic peptide XP1 has been used for the production of antibodies.

Author Response

Responses to reviewers suggestions.

1) The numbering on Figure 4A has been corrected in a revised version (Figure 4 R3)

2) The peptide sequences given in part A of Fig S1 R2 were erroneous and have been corrected in a revised version (Fig S1 R3).

3) Consequently, the correct sequence of peptide XP1 is now shown in both Figure 4A and in Figure S1A

We thank the reviewer for their useful comments.